# Modulation of Melatonin in Pain Behaviors Associated with Oxidative Stress and Neuroinflammation Responses in an Animal Model of Central Post-Stroke Pain

**DOI:** 10.3390/ijms24065413

**Published:** 2023-03-12

**Authors:** Tavleen Kaur, Andrew Chih-Wei Huang, Bai-Chuang Shyu

**Affiliations:** 1Taiwan International Graduate Program in Interdisciplinary Neuroscience, National Yang-Ming Chiao Tung University and Academia Sinica, Taipei 11529, Taiwan; tavleenkaurisf@gmail.com; 2Division of Neuroscience, Institute of Biomedical Sciences, Academia Sinica, Taipei 11529, Taiwan; 3Department of Psychology, Fo Guang University, Yilan County 26247, Taiwan; chweihuang@mail.fgu.edu.tw

**Keywords:** central poststroke pain, neuromodulation, melatonin, inflammation, mitochondrial dysfunction

## Abstract

Central post-stroke pain is a severe persistent pain disease that affects 12% of stroke survivors (CPSP). These patients may have a cognitive impairment, depression, and sleep apnea, which leave them open to misdiagnosis and mistreatment. However, there has been little research on whether the neurohormone melatonin can effectively reduce pain in CPSP conditions. In the present study, we labeled melatonin receptors in various brain regions of rats. Later, we established a CPSP animal model by intra-thalamic collagenase lesions. After a rehabilitation period of three weeks, melatonin was administered using different doses (i.e., 30 mg/kg, 60 mg/kg, 120 mg/kg) for the following three weeks. Mechanical allodynia, thermal hyperalgesia, and cold allodynia behavioral tests were performed. Immediately after behavioral parameters were tested, animals were sacrificed, and the thalamus and cortex were isolated for biochemical (mitochondrial complexes/enzyme assays and LPO, GSH levels) and neuroinflammatory (TNF-α, IL-1β, IL-6) assessments. The results show that melatonin receptors were abundant in VPM/VPL regions. The thalamic lesion significantly induced pain behaviors in the mechanical, thermal planters, and cold allodynia tests. A significant decrease in mitochondrial chain complexes (C-I, II, III, IV) and enzymes (SOD, CAT, Gpx, SDH) was observed after the thalamic lesion. While there were significant increases in reactive oxygen species levels, including increases in LPO, the levels of reduced GSH were decreased in both the cortex and thalamus. Proinflammatory infiltration was noticed after the thalamic lesion, as there was a significant elevation in levels of TNF-α, IL-1β, and IL-6. Administration of melatonin has been shown to reverse the injury effect dose-dependently. Moreover, a significant increase in C-I, IV, SOD, CAT, and Gpx levels occurred in the CPSP group. Proinflammatory cytokines were significantly reduced by melatonin treatments. Melatonin seems to mediate its actions through MT1 receptors by preserving mitochondrial homeostasis, reducing free radical generation, enhancing mitochondrial glutathione levels, safeguarding the proton potential in the mitochondrial ETC by stimulating complex I and IV activities, and protecting the neuronal damage. In summary, exogenous melatonin can ameliorate pain behaviors in CPSP. The present findings may provide a novel neuromodulatory treatment in the clinical aspects of CPSP.

## 1. Introduction

Central post-stroke pain (CPSP) is neuropathic pain due to associated lesions of the central somatosensory nervous system. This syndrome is characterized by pain and sensory abnormalities in the body parts corresponding to brain regions injured by the cerebrovascular lesion [1]. Patients may also develop spontaneous pain with burning, aching, squeezing, pricking, numb sensations, and sensory disturbances such as dysesthesia, hyperalgesia, and allodynia due to the non-nociceptive stimulus. It has previously been shown that, out of 166 stroke survivors, 8% of patients suffered from CPSP symptoms. Moreover, younger age groups (<60 years) are significantly associated with CPSP. Pain characteristics of CPSP include hyperesthesia (10.71%), electric shocks (9.64%), temperature allodynia (9.64%), and allodynia (12.86%) [2]. Current treatments for post-stroke pain include anti-convulsant drugs, opioids, antidepressants, N-methyl-D-aspartate (NMDA) antagonists, and motor cortex stimulation; these anti-pain medicines are primarily developed to reduce neuronal hyperexcitability in peripheral and central nervous systems. The adverse effects of these medicines are often associated with worse side-effects and a high probability of drug addiction. CPSP symptoms are challenging to ameliorate because of their severity, chronicity, and resistance to simple analgesics [1].

Mitochondria, often referred to as the powerhouses of the cell, are present in nearly all types of human cells. The ability of cells to generate energy from ambient oxygen is facilitated by the role of mitochondria as energy providers and signaling intermediaries. The ETC transforms metabolic substrate electrons into molecular oxygen (O_2_), creating an electrochemical gradient of H+ whose energy produces ATP [3]. Mitochondria play a significant role in critical events such as cell death, cell survival, and neurodegeneration [4]. Mitochondrial dysfunction is a causal or contributory mechanism for normal sensory processing and chronic pain. Much evidence involves different aspects of mitochondria and their functions in human and animal pain conditions, addressing topics such as mitochondrial ultrastructure, distribution, oxygen consumption, oxidative phosphorylation, calcium buffering, and ROS and ATP levels [5]. Furthermore, mitochondrial function abnormalities, such as defects in the electron transport chain (ETC)/oxidative stress and ATP production, underlie neurodegenerative diseases of different etiologies, such as Parkinson’s disease (PD), Alzheimer’s disease (AD), and Huntington’s disease (HD) [6,7,8,9,10]. This raises the question: could there be mitochondrial deficits in neuropathic pain conditions such as CPSP as well? Therefore, this study assessed the mitochondrial function to determine the possible effects of mitochondria in the CPSP condition.

To our knowledge, melatonin, called N-acetyl-5-methoxy tryptamine, is a neuro-hormone. SCN regulates the biosynthesis of melatonin in the pineal gland, and it contributes to synchronization in the light–dark cycle of the environment [11]. It acts through its MT1/MT2 receptors in mammals and has also been found to be involved in non-receptor-mediated functions [12,13]. Alternatively, melatonin has also been shown to regulate an individual’s circadian rhythms and mood alternation. Furthermore, melatonin plays a pivotal role in pain regulation. The administration of melatonin can be served as an analgesic and anxiolytic agent and can, therefore, produce ameliorative effects in fibromyalgia, migraine, and irritable bowel syndrome [14]. Previously, some studies have reported that melatonin exhibits neuroprotective effects in animal models of brain injury, such as SCI and TBI [15,16]. Furthermore, selective MT2 receptor partial agonists achieve analgesic properties through the modulation of brainstem-descending anti-nociceptive pathways [17]. Additionally, multiple studies have previously reported that melatonin effectively regulates mitochondrial homeostasis [18,19]. Melatonin, because of its amphiphilic nature, crosses cell membranes easily, enters body fluids easily, and accumulates within subcellular compartments such as mitochondria [20]. A previous study has reported that melatonin can stabilize the fluidity of mitochondrial inner membranes [21]. Another previous study using 2-[125I]-iodomelatonin reported that melatonin binds to mitochondrial membranes [22,23].

Therefore, the present study addressed the following issues: first, whether melatonin could ameliorate pain behavior as a clinical intervention in the CPSP condition; second, whether the antinociception of melatonin was associated with oxidative stress responses, mitochondrial ETC enzymes and complexes function, or neuroinflammation.

## 2. Results

### 2.1. Location Assessments of Melatonin Receptors

The distribution of melatonin receptors was assessed in the various brain regions, including the ventral posteromedial thalamic nucleus (VPM, Figure 1A), somatosensory cortex (S1, Figure 1B), dentate gyrus (DG, Figure 1C), amygdalostriatal transition area (Astr, Figure 1D), secondary auditory cortex (AUD, Figure 1E), basolateral amygdala (BLA, Figure 1F), lateral hypothalamus (LH, Figure 1G), magnocellular region (Mc, Figure 1H), magnocellular nucleus of the lateral hypothalamus (Mclh, Figure 1I), nigrostriatal fibers (Ns, Figure 1J), posterior hypothalamic area (Ph, Figure 1K), retrosplenial agranular cortex (Rsa, Figure 1L), posterior thalamic nuclear group (Po., Figure 1M), ventromedial thalamic nucleus (VM, Figure 1N), ventromedial hypothalamic nucleus (VMH, Figure 1O), and reticular thalamic nucleus (RT, Figure 1P). The receptor-mediated actions of melatonin might contribute to functions of the central nervous system, especially in pain behaviors. A distribution of melatonin receptor MT1 was found in various brain areas (Table 1). The maximum count of receptor distribution was found in the somatosensory cortex S1, hippocampus, ventral posterior medial/lateral nucleus VPM/VPL, and auditory cortex AUD (see Figure 1 and Table 1).

### 2.2. Induction of CPSP and Pain Behaviors

To assess pain sensitivity, pain behavioral tests were conducted on sham control, lesion control, lesion + MLT (30 mg/kg), lesion + MLT (60 mg/kg), and lesion + MLT (120 mg/kg) groups. A Kruskal–Wallis analysis indicated that there were significant differences among all groups for 7D (Kruskal–Wallis test = 20.79, *p* < 0.05) and 14D (Kruskal–Wallis test = 20.08, *p* < 0.05). The mechanical hyperalgesia threshold in the Von Frey test was significantly decreased due to the lesion of the VPM/VPL, suggesting that it caused the CPSP pain behavior. However, 120 mg/kg of melatonin administration (but not the other doses of melatonin) significantly increased the mechanical hyperalgesia threshold, indicating that 120 mg/kg of melatonin administration reduced pain perception compared with the lesion control group for 7D (*p* < 0.05) and 14D (*p* < 0.05, Figure 2A). The results suggest that the higher dose of melatonin was more effective in reducing mechanical pain perception.

Concerning the cold allodynia test, a Kruskal–Wallis test indicated that there was a significant difference in the factor of the group (Kruskal–Wallis test = 31.13, *p* < 0.05). However, the post hoc with the Dunn test indicated that there was no significant difference in the CPSP compared with the sham control group, possibly because of the high variance in the cold allodynia scores (*p* > 0.05; Figure 2B); this indicates that measurement of the cold allodynia did not show pain behaviors in the animal model of CPSP.

Furthermore, the plantar test showed significant differences occurring in groups 7D (Kruskal–Wallis test = 19.16, *p* < 0.05) and 14D (Kruskal–Wallis test = 20.10, *p* < 0.05), indicating that the rats in the lesion group had a shorter paw withdrawal time for 7D and 14D compared with the sham control group (*p* < 0.05). Moreover, the lesion + MLT (60 mg/kg) and lesion + MLT (120 mg/kg) groups had a significantly increased paw withdrawal time compared with the lesion group for 7D and 14D (*p* < 0.05). Thus, the 60 mg/kg and 120 mg/kg melatonin administrations (but not the 30 mg/kg melatonin group) effectively reduced thermal hyperalgesia after 7D and 14D melatonin administrations (Figure 2C).

### 2.3. Determining Whether Melatonin Can Prevent Oxidative Stress Caused by Mitochondrial Dysfunction in CPSP Animal Model

The oxidative stress parameters and mitochondrial antioxidant levels were assessed to verify the dose-dependent effect of melatonin (i.e., 30, 60, and 120 mg/kg) on oxidative stress in the 7 + 14 days (after 3 weeks of melatonin administration) following injury and rehabilitation.

#### 2.3.1. Effect of Lesions on Electron Transport Chain Complexes and the Effect of Melatonin Treatment

The four mitochondrial chain complex levels were evaluated in both the thalamus and cortex tissues.

The levels of complex-i were found to be decreased in lesioned animals in both thalamic and cortical tissues. However, there was no effect of 30 mg/kg and 60 mg/kg melatonin treatments in cortical tissues. Only the highest dose of 120 mg/kg was significantly effective, while all three doses of melatonin significantly enhanced levels of complex-i in the thalamus (Kruskal–Wallis test = 22.39, *p* < 0.05) (Figure 3A).

Furthermore, there were no significant differences in levels of complex-ii in the thalamus among all groups (Kruskal–Wallis test = 8.91, *p* > 0.05), indicating that the complex-ii levels were not involved in the thalamus for the CPSP animals (Figure 3B). In the cortex, a significant difference occurred in the group in the levels of complex-ii (Kruskal–Wallis test = 17.80, *p* < 0.05). Post hoc Dunn tests showed that the complex-ii levels were significantly decreased in the CPSP control group compared with the sham control (*p* < 0.05). Compared with the CPSP control group, CPSP + MLT (60 mg/kg) and CPSP + MLT (120 mg/kg) were significantly decreased in the complex-ii levels in the cortex (Figure 3B).

The levels of complex-iii were robustly decreased in both the thalamus (Kruskal–Wallis test = 14.20, *p* < 0.05) and the cortex (Kruskal–Wallis test = 11.91, *p* < 0.05); however, melatonin treatments did not affect complex-iii levels in the thalamus and cortex (*p* > 0.05; Figure 3C).

To test complex-iv levels in the thalamus and cortex, a Kruskal–Wallis test was used, indicating that there were significant group differences in the thalamus (Kruskal–Wallis test = 22.08, *p* < 0.05) and cortex (Kruskal–Wallis test = 20.61, *p* < 0.05). The Dunn post hoc indicated that the CPSP group had significantly decreased complex-iv levels in the thalamus (*p* < 0.05) and cortex (*p* < 0.05) compared with the sham control group. Melatonin treatments in 30~120 mg/kg doses could increase complex-iv levels in the thalamus and cortex (*p* < 0.05; Figure 3D).

Therefore, our results suggested that melatonin was significantly effective in rescuing levels of complex-i and complex-iv. Melatonin treatments significantly decreased the complex-ii levels in the cortex for the CPSP animal.

#### 2.3.2. Effect of Lesions on Electron Transport Chain Enzymes and the Effect of Melatonin Treatment

To assess the effect of lesion and melatonin treatments on electron transport chain enzymes, we used a Kruskal–Wallis test that indicated that there were significant group differences in the levels of glutathione peroxidase (GPx) in the thalamus (Kruskal–Wallis test = 18.14, *p* < 0.05) and the cortex (Kruskal–Wallis test = 17.39, *p* < 0.05). The Dunn post hoc indicated that the lesions reduced GPx levels in the thalamus (*p* < 0.05) and the cortex (*p* < 0.05); moreover, melatonin treatments increased GPx levels in the thalamus (*p* < 0.05) and the cortex, particularly under 120 mg/kg doses of melatonin (*p* < 0.05; Figure 4A).

Furthermore, levels of catalase were reduced significantly after CPSP in both the thalamus (Kruskal–Wallis test = 17.67, *p* < 0.05) and the cortex (Kruskal–Wallis test = 18.36, *p* < 0.05) (Figure 5B). However, treatments with 30–120 mg/kg doses of melatonin may have increased catalase levels in the thalamus (*p* < 0.05) and the cortex (*p* < 0.05; Figure 4B).

Furthermore, after CPSP, the CPSP animals demonstrated significantly decreased superoxide dismutase levels in the thalamus (Kruskal–Wallis test = 22.07, *p* < 0.05) and the cortex (Kruskal–Wallis test = 22.21, *p* < 0.05) compared with the sham control group. However, 30–120 mg/kg doses of melatonin treatments significantly increased superoxide dismutase levels in the thalamus (*p* < 0.05) and the cortex (*p* < 0.05; Figure 4C).

#### 2.3.3. Effect of Lesion and Melatonin Treatments on Oxidative Stress

The hallmark of mitochondrial oxidative stress is the presence of ROS, e.g., OH– (hydroxyl radical), O_2_–(superoxide radical), and H_2_O_2_ (hydrogen peroxide), which leads to mitochondrial dysfunction. Therefore, we assessed levels of MDA and levels of reduced glutathione. Robust increases in levels of MDA after CPSP were observed in both thalamus (Kruskal–Wallis test = 22.15, *p* < 0.05) and cortical tissues (Kruskal–Wallis test = 23.08, *p* < 0.05), suggesting enhanced lipid peroxidation in the thalamus and the cortex after CPSP. However, significant reductions in MDA levels were observed under 30–120 mg/kg doses of melatonin in the thalamus (*p* < 0.05) and the cortex (*p* < 0.05; Figure 5A).

The levels of reduced glutathione after CPSP were significantly decreased in the thalamus (Kruskal–Wallis test = 21.44, *p* < 0.05) and cortex (Kruskal–Wallis test = 23.08, *p* < 0.05), suggesting the presence of oxidative stress in these brain areas. While treatment with 30 mg/kg had no effect on thalamic tissue (*p* > 0.05), both 60 mg/kg and 120 mg/kg treatments were associated with significant improvements in levels of reduced glutathione in the thalamus (*p* < 0.05) and the cortex (*p* < 0.05; Figure 5B).

### 2.4. Effect of Melatonin on Cytokine Levels

In determining neuroinflammation in both CPSP rats and melatonin-treated rats, a one-way ANOVA was used. Neurodegeneration due to stroke leads to mitochondria-related oxidative stress, which may activate microglia, thereby triggering the infiltration of neuro-inflammatory cytokines such as TNF-α (Kruskal–Wallis test = 21.52, *p* < 0.05) (Figure 6A), IL-6 (Kruskal–Wallis test = 20.30, *p* < 0.05) (Figure 6B), and IL-1b (Kruskal–Wallis test = 24.04, *p* < 0.05) (Figure 6C) into the whole brain tissue. There was a significant increase in neuroinflammatory cytokines in the lesion animals in the whole brain compared with the sham control group (*p* < 0.05). Treatments with melatonin in 60 mg/kg and 120 mg/kg doses significantly reduced IL-1b levels, suggesting that higher doses of melatonin of 60–120 mg/kg could ameliorate neuroinflammation responses.

## 3. Discussion

In previous studies, melatonin receptors have been identified using RT-PCR in various tissues in humans and rats, such as in the cerebellum, SCN, entorhinal cortex, pars tuberalis, pineal gland, neurohypophysis, hypothalamus, bone marrow, blood, and spleen [24]. Similarly, in our study, we found an evaluated abundance of melatonin receptors in various regions, viz., S1, Vm, RT, hippocampus, MePD/MePV, VPM, VPL, lateral hypothalamus, VMHYP, AUD, PH, BLA, and S2 of rats. VPM/VPL, AUD, and S1 regions had the highest count of melatonin receptors, suggesting that exogenous melatonin was pertinent in our study.

Collagenase microinjections can be successfully induced in CPSP, indicating decreases in the withdrawal threshold and showing mechanical hyperalgesia. The quick paw withdrawal response also decreased with cold allodynia, and there was a decrease in paw withdrawal latency in plantar tests; additionally, melatonin alleviated the pain caused by CPSP in all pain behavior tests. However, the effect of melatonin was shown to be dose-dependent, with data suggesting that higher doses of 120 mg/kg have a more significant effect than lower (30 mg/kg) and medium (60 mg/kg) doses.

Moreover, our previous study showed reduced levels of endogenous melatonin due to CPSP [25]. Some other studies have previously reported that methionine synthase affects the secretion of endogenous melatonin in the pineal gland [26]. The reduced levels of endogenous melatonin could be due to blockage of methionine synthase, which, in turn, could be due to self-perpetuating, vicious cycles of oxidative stress radicals. Hence, reduced levels of endogenous melatonin and oxidative stress may be linked. When the formation of free radicals outweighs their removal, it is said to be under oxidative stress, which increases the amount of oxidative damage to biomolecules. In our work, we discovered that this occurred after CPSP, with greater levels of free radical production and weakened antioxidant defenses resulting in increased production of OH, a strong precursor to lipid peroxidation.

Tissue damage due to stroke may cause mitochondrial dysfunction, thereby inducing an oxidative stress cascade and apoptosis and leading to mitochondrial membrane permeabilization (MMP). This may give rise to matrix calcium levels causing electron transport chain (ETC) dysfunction in mitochondria and ETC failure, and lead to the generation of reactive oxygen and reactive nitrogen species by mitochondria (Figure 7).

Our results showed that melatonin treatments had a significant effect on mitochondrial function and are consistent with previous reports. However, melatonin has shown protective effects in various diseases, such as AD and PD, in which mitochondrial dysfunction is one of the causes of the condition [27]. The mechanism by which melatonin shows a neuroprotective effect seems to be caused by its anti-oxidant and free-radical scavenging properties. Additionally, in our study, melatonin showed a significant effect on enhancing the activities of complex I and complex IV on the mitochondrial electron transport chain, suggesting that melatonin might interact with complexes of the electron transport chain and act as an oxidized intermediate as a melatonin cation, accepting electrons that, in turn, increase the electron flow, thereby supporting the net electron flux, proton potential, and ATP synthesis in mitochondria [20].

The brain is particularly susceptible to damage under oxidative stress because it is rich in phospholipids and proteins that are susceptible to oxidative damage and may not have significant quantities of antioxidant defense enzymes. Free-radical-mediated injury may play an important role in the severity of the disease. Superoxide Dismutase (SOD) down-regulation marks the amount of increased superoxide free radicals in mitochondria, while catalase (CAT) is important for the protection of cells from oxidative damage by ROS and catalyzes the breakdown of hydrogen peroxide to water and oxygen.

The hallmark of mitochondrial oxidative stress is the presence of ROS, e.g., OH– (hydroxyl radical), O_2_– (superoxide radical), and H_2_O_2_ (hydrogen peroxide), which lead to mitochondrial dysfunction. In our study, melatonin restored the decreased levels of antioxidants SOD and CAT due to CPSP. Thus, melatonin has shown remarkable antioxidant properties in scavenging hydroxyl, superoxide, and hydrogen peroxide. We suggest that mechanisms that terminate the radical reaction chains involve the combination and detoxification of superoxide anions with melatonin radicals produced as a result of scavenging.

In addition, melatonin administration decreased lipid peroxidation (LPO) by interacting with lipid bilayers, stabilizing mitochondrial inner membranes, and restoring oxidative damage by augmenting GSH levels and glutathione peroxidase (GPx) activities. Hence, it may improve the ETC activity of mitochondria by safeguarding the electron flow and thereby increasing ATP production.

Oxidative stress cascades produce reactive oxygen or nitrogen species (ROS/RNS), leading to mitochondrial dysfunction. This oxidative stress cascade causes lipid peroxidation and activates p38MAPK and, thus, causes the high activity of the NFκB transcription factor and, in turn, increased proinflammatory cytokines such as TNF-α and IL-6; this cycle continues [28]. Damage to the neurons and inflammation may cause abnormal hyper-excitability in the nociceptive sensory neurons, such that even a nonpainful stimulus could cause subthreshold firing of nociceptive neurons, contributing to central post-stroke pain. Administration of melatonin may activate the ERK1/2 pathway, thereby demonstrating antinociception by decreasing mitochondria-derived ROS and reducing proinflammatory cytokines, and thus alleviating pain in CPSP animal models. In summary, how melatonin, through the signaling pathways of MAPK or NFκB, modulates oxidative stress and inflammation due to changes in central post-stroke pain should be evaluated in further studies.

However, our study has certain limitations. We did not assess the role of melatonin in opiate and dopaminergic signaling. SCN has melatonin, opioid, and dopamine receptors, which may interfere with melatonin’s antinociception signaling. Thus far, melatonin has been found to have no adverse effects. Therefore, exogenous melatonin administration in chronic pain conditions may be a promising clinical intervention for drug tolerance and addiction.

## 4. Materials and Methods

### 4.1. Animals and Drugs

Twenty-five male Sprague Dawley rats (~8 weeks of age) were purchased from BioLASCO, Yi-Lan, Taiwan. They were housed (one rat per cage) in an animal room under a 12 h light–dark cycle with air conditioning and 60% humidity and received a chow diet and water ad libitum. All experiments were performed in accordance with the guidelines of the Academia Sinica Institutional Animal Care and Utilization Committee.

### 4.2. Experimental Procedure

The experimental procedure is shown in Figure 8A. At the beginning of the experiment, animals were allowed habituation and pre-training from days 0 to 6. On day 7, the lesion was induced. Then, there was a rehabilitation period of three weeks following the lesion. After three weeks following the rehabilitation period, melatonin was administrated. Behavior tests were conducted for the baseline and lesioned animals and for the melatonin treatment (after the first 7 days of melatonin treatment, 7D; after the next 14 days of melatonin treatment, 14D; see Figure 8A). The surgery was performed with the stereotaxic instrument; the successful induction of lesion VPA and VPL is shown in Figure 8B.

### 4.3. CPSP Induction

CPSP was induced by thalamic lesions, and the procedures were performed using the method described by Kuan et al. in 2015 [29]. During surgery, the animals were kept under 1% isoflurane anesthesia. Using a homeothermic blanket system, the body’s temperature was maintained between 36.5 and 37.5 °C (Model 50-7079, Harvard Apparatus, Holliston, MA, USA). Type IV collagenase (C5138, SIGMA, Saint Louis, MO, USA; 0.125 U/0.5 L saline) was administered intravenously to the animals and injected into the right ventral posterior medial nucleus (VPM)/ventral posterior lateral nucleus (VPL) of the thalamus (coordinates: 3.0–3.5 mm posterior, 3.0–3.4 mm lateral to bregma, 5.7–6.0 mm depth). Further experiments were conducted after the three-week post-surgery rehabilitation period.

### 4.4. MT1 Receptor Distribution in CNS: Using Immunohistochemistry

At −20 °C, frozen tissue samples were sliced into 10 μm thick cryo-sections using a cryostat microtome. The non-specific antibody binding was blocked with 3% bovine serum albumin (Sigma-Aldrich, St. Louis, MO, USA) and 0.3% Triton X-100 in phosphate-buffered saline (PBS, pH 7.4) for 30 min. The sections were incubated in the primary antibody solution (1:100 dilution, MT1, Mel 1a Receptor, MTNR1A: AMR-031, Alomone labs, Israel) for 30 min. Then, the sections were incubated in secondary antibody solution (1:1000 dilution, Alexa Fluor 488-labeled goat anti-rabbit IgG, Thermo Fisher Scientific, 29851 Willow Creek Road, Eugene, OR, USA.) for 1 h and counterstained with DAPI for 3 min. Except where otherwise noted, all dilutions and thorough washings between phases were carried out using PBS. The images of the sections were obtained using a Pannoramic 250 FLASH II slide scanner, and the total count of receptor distribution in different regions of rat brains was performed using Zen software on panoramic images.

### 4.5. Pain Behavior Test

#### 4.5.1. Von Frey

The mechanical pain behavior test was performed by the Von Frey device. The animals were put on an elevated mesh and given 30 min to explore. A specified force was applied while maintaining compression with an elastic filament. The diameters of filaments require a wide range of forces to induce buckling. The shorter the filament, the higher the force required to buckle it. It helped to record the minimal force/pressure at which the animals reacted (limb withdrawal) due to the painful stimulus. Each hind limb underwent three trials, and the average of the minimal pressure was used to determine the threshold. A 5 min break separated every trial.

#### 4.5.2. Plantar Test

The Thermal Plantar Instrument measures pain sensitivity according to Hargreave’s Method (IITC Inc. Life Science, 23924 Victory Blvd Woodland Hills, CA 91367, USA). It measures the infrared heat stimulus response in rodents and functions by focusing the infrared source below the plexiglass surface (instrument) and pressing the button of the instrument. Rats would then be placed on a plexiglass surface for about 30 min. The radiant heat would be focused on the hind paw below the surface of the glass floor; paw withdrawal latency and infrared intensity are recorded automatically. Each rat was tested three times for each hind paw, with a 5 min interval between tests.

#### 4.5.3. Cold Allodynia

The procedure of cold allodynia was performed in accordance with the method described by Flatters and Bennett [30]. Animals were placed on a wired mesh floor and several drops of acetone were poured onto the ventral surface (plantar) of the left and right hind paws. The cut-off time was 30 s. The response to the cooling effect of acetone was recorded as scores. Paw withdrawal was recorded as quick withdrawal = 1; prolonged response = 2; repeated flicking and ventral licking of paw = 3, and no response = 0. The response for each paw was recorded three times. The scores were added to obtain a cumulative response.

### 4.6. Experimental Procedures for Evaluating Oxidative Stress and Mitochondrial Function

#### 4.6.1. Dissection and Homogenization

After the behaviors, the animals were sacrificed. After brain extraction from the rat’s skull, the brain was rinsed with cold PBS to remove excess blood. Then, the tissues were isolated on dry ice with forceps and scissors. Because the previous studies demonstrated that the thalamus and cerebral cortex contributed to chronic pain [31,32], the whole cerebral cortex and whole thalamus were determined for assessments. Later, brain samples were quickly taken out and put on dry ice to isolate the cortex and thalamus. Then, 10% (*w*/*v*) tissue homogenates were made in 0.1 M phosphate buffer (pH 7.4). The homogenates were centrifuged at 10,000× *g* for 15 min. Aliquots of supernatants were separated and utilized for biochemical and molecular estimations.

#### 4.6.2. Isolation of Mitochondria

Brain tissues were homogenized in 230 mM mannitol, 70 mM sucrose, 1.0 mM EDTA, and 10 mM Tris-HCl, pH 7.40, at a ratio of 10 mL of homogenization medium/g of tissue, and mitochondria were extracted from these tissues. To obtain mitochondrial preparations, the homogenate was centrifuged at 700× *g* for 10 min and the supernatant was centrifuged at 8000× *g* for 10 min to pellet the mitochondria that were then washed under the same circumstances [33]. Once collected, the mitochondria were maintained in the same medium. The succinate dehydrogenase activity was used to gauge the quality of the mitochondrial preparation.

#### 4.6.3. Protein Estimation

Protein content was measured with the Lowry method using a Folin phenol reagent [34]. The phenolic group of tyrosine and tryptophan residues (amino acid) in a protein produces a blue-purple color complex with the Folin–Ciocalteau reagent, which consists of sodium tungstate molybdate and phosphate. Absorbance was measured at 660 nm using a plate reader.

Preparing 0.1 mL of the sample or standard (10 μg, 20 μg, 40 μg, 80 μg, and 100 μg/mL), 0.1 mL of 2 N NaOH was added and hydrolyzed at 100 °C for 10 min in a heating block or boiling water bath. Hydrolysate was allowed to cool down, and 1 mL of freshly mixed complex-forming reagent (2% (*w*/*v*) Na_2_CO_3_ in distilled water + 1% (*w*/*v*) CuSO_4_·5H_2_O in distilled water + 2% (*w*/*v*) sodium potassium tartrate in distilled water) was added to it. The solution was maintained at room temperature for 10 min. Then, 0.1 mL of Folin reagent was added to the above solution, and the mixture was maintained at room temperature for 30–60 min (did not exceed 60 min). Absorbance was noted at 660 nm.

#### 4.6.4. Estimation of Oxidative Stress Markers: Measurement of Lipid Peroxidation

Malondialdehyde (MDA), an end product of lipid peroxidation, was measured quantitatively in the brain [35]. During lipid oxidation, MDA, a minor component of fatty acids with three or more double bonds, is formed as a result of the degradation of polyunsaturated fatty acids. It is frequently utilized as a lipid oxidation process indicator due to its early occurrence as oxidation takes place and the analytical method’s sensitivity. This experiment’s MDA and thiobarbituric acid (TBA) reacted to generate a pink MDA-TBA complex, seen at 532 nm. After that, 0.5 mL of Tris HCl was added to 0.5 mL of supernatant and incubated at 37 °C for 2 h. After incubation, 1 mL of 10% trichloroacetic acid (TCA) was added and centrifuged at 10,000× *g* for 10 min. To 1 mL of supernatant, 1 mL of 0.067% thiobarbituric acid was added and the tubes were maintained in boiling water for 10 min. After cooling, 1 mL of double-distilled water was added and the amount of malondialdehyde (MDA), a measure of lipid peroxidation, was measured by reaction with thiobarbituric acid at 532 nm.

#### 4.6.5. Estimation of Oxidative Stress Markers: Estimation of Reduced Glutathione

The amino acids glutamine, cysteine, and glycine are combined to form the water-soluble tripeptide known as glutathione (GSH). GSH, which can reach millimolar quantities in some tissues, is the most prevalent intracellular small-molecule thiol due to the thiol group’s powerful reducing agent. Glutathione S-transferases (GST) and glutathione peroxidases catalyze the detoxification of a variety of electrophilic substances and peroxides using GSH as an essential antioxidant (GPx). 5,5′-dithio-bis(2-nitrobenzoic acid), often known as DTNB and Ellman’s [36], was first made available in 1959. It is a substance that dissolves in water and is used to gauge the concentration of soluble sulfuryl groups (-SH) in a solution. A yellow color is created when DTNB interacts with the sulfhydryl group. Thus, using the technique outlined by Ellman (1959) [37], the level of reduced glutathione in the cortex and thalamus of brain samples was calculated. The concentration of glutathione was represented as μmol per mg protein, and the absorbance was measured at 412 nm.

A mixture of homogenate and 4% sulphosalicylic acid was maintained at 4 °C for one hour before being centrifuged at 1200× *g* for 15 min at 4 °C. The supernatant was collected, and phosphate buffer (0.1 M, pH 8) and DTNB (0.4% *w*/*v* in 0.1 M phosphate buffer, pH 8) were added to it. At 412 nm, absorbance was immediately measured.

#### 4.6.6. Estimation of Mitochondrial Complexes and Enzymes

Complex—I: Also known as NADH: ubiquinone oxidoreductase, Type I NADH dehydrogenase, and mitochondrial complex I. It was assayed with the method described by King and Howard [38]. The various reagents used were: 0.2 M glycyl glycine (168.6) = 335 mg/10 mL (pH 8.5); 6 mM NADH (709.4) = 42.6 mg/10 mL in glycyl glycine buffer; 1.05 mM cytochrome-C = 13.65 mg/mL; 0.02 M sodium bicarbonate = 16.8 mg/10 mL. Method: 100 µL Cyct-C (50 µL) was added to the solution of 350 µL of glycyl glycine buffer (175 µL), and then 100 µL of NADH (50 µL), 2.4 mL of DDW (1.2 mL), 10 µL of the sample (5 µL) and 20 µL of NaHCO_3_ (10 µL) were added.

Complex—II: Also known as succinate dehydrogenase (SDH); it was assayed with the modified method of King (1967) [39]. The various reagents used were: 100 mL of sodium phosphate buffer (0.2 M; pH 7.8), where NaH_2_PO_4_ = 3.12 g in 100 mL and Na_2_HPO_4_ = 2.83 g in 100 mL; BSA (1%) = 100 mg/100 mL (unshaken); Succinic acid (freshly prepared) 0.6 M—700 mg/10 mL DDW; Potassium ferricyanide 0.03 M (freshly prepared), 9.8 mg/1 mL. The procedure was performed as follows: 200 µL of succinate was added to 1.5 mL of phosphate buffer, and then 300 µL of BSA, 25 µL of Pot. Ferricyanide, 1.75 mL of DDW, and 25 µL of the sample were added, and the change in OD was checked at 420 for 180 min.

Complex—III: This complex is also known as Coenzyme Q—cytochrome c reductase. MTT, 10 mg/mL, in 0.1 PBS and DMSO were used as reagents [40]. An amount of 10 µL of MTT was added to 100 µL of the sample and incubated on a plate for 3 h at 37 °C. Later, 200 µL of DMSO was added and read at 580 in the ELISA reader.

Complex—IV: This complex is also known as Cytochrome c oxidase. This assay was performed using the method described by Sottocasa (1967) [41]. The reagents used were 100 mL of sodium phosphate buffer (0.075 M; pH 7.4), where NaH_2_PO_4_ = 1.17 g in 100 mL and Na_2_HPO_4_ = 1.06 g in 100 mL; 0.3 mM cytochrome C (reduced): 3.9 mg/mL in phosphate buffer; 100 mM HCl at 0.43 mL/50 mL DDW, and several crystals of sodium borohydride to reduce Cyto-C.

Cyto-C was reduced by the addition of several crystals of sodium borohydride (light-sensitive) and neutralized with 100 mM HCl (pH 7). Then, 700 µL of phosphate buffer was added to the solution of 100 µL of reduced Cyto-C (50 µL) and 10 µL of the sample and the change in OD was checked at 550 for 180 min.

##### Estimation of Catalase (CAT) Activity

The estimation of catalase was performed with the method described by Luck [42]. The reagents used were: 50 mM Phosphate buffer (pH 7.0) and 12.5 mM H_2_O_2_ in phosphate buffer (0.16 mL of diluted H_2_O_2_ (30% *w*/*v*) to 100 mL with phosphate buffer (protected from light). Method: Test: 3 mL of H_2_O_2_-phosphate buffer + 0.05 mL of supernatant; Blank: 0.05 mL of supernatant + phosphate buffer (3 mL). Change in OD measured at 240 nm for 2 min with 30/60 s intervals.

##### Determination of Glutathione Peroxidase (GPx) Activity

The Ellman method was used to carry out the assay [36]. The reagents used were 4 mM EDTA: 14.88 mg/10 mL water, Ellman’s reagent: 19.8 mg of DTNB (dithionitrobis benzoic acid) in 1% sodium citrate; 0.4 M Phosphate buffer (pH 7.0); 10 mM Sodium azide (NaN3); 2.5 mM Hydrogen peroxide (H_2_O_2_); 4 mM reduced glutathione; 10% Trichloroacetic acid (TCA); 0.3 M disodium hydrogen phosphate solution. Method: Two test tubes marked test (T) and control (C) received 0.4 mL of buffer, 0.2 mL of EDTA, 0.1 mL of NaN3, 0.2 mL of reduced glutathione, and 0.1 mL of H_2_O_2_ (C). An amount of 0.2 mL of the sample was introduced to the test, and 0.2 mL of water was placed in the control. After thoroughly mixing the materials, they were incubated at 37 °C for 10 min. By adding 0.5 mL of 10% TCA, the reaction was stopped, and the mixture’s contents were centrifuged. The color generated was then read at 412 nm after 3.0 mL of buffer and 0.5 mL of Ellman’s reagent were added to 1.0 mL of the supernatant. Standards in the range of 40–200 μg of reduced glutathione were taken and treated similarly. The activity was expressed in terms of μmoles of glutathione consumed/min/mg protein or mL of serum.

##### Estimation of Superoxide Dismutase (SOD) Activity

The enzyme superoxide dismutase was assayed using the modified method by Kono et al. [43]. In this biochemical method, nitroblue tetrazolium (NBT) reduction was used as an indicator of O_2_ production. SOD competed with NBT for O_2_, and the percent inhibition of NBT reduction is a measure of the amount of SOD that was present. In this method, 50 mM Na_2_CO_3_ (0.52 g/100 mL) in 0.1 mM EDTA (0.03 g/100 mL) (pH 10.8) solution was prepared, while 96 mM NBT (nitroblue tetrazolium) was dissolved in this solution (0.008 g/100 mL). Then, 2 mL of this NBT solution was added to 0.5 mL of hydroxylamine HCl (20 mM Hydroxylamine HCl with adjusted pH of 6 with NaOH) to generate controls. For tests, 2 mL of NBT solution was added to 0.5 mL of hydroxylamine HCl, and 0.1 mL of PMS of homogenate was added. Later, the change in OD at 560 nm was measured for 2 min at 30/60 s intervals.

#### 4.6.7. Estimation of Pro-Inflammatory Cytokines (IL-1β, IL-6, and TNF-α) Levels

Rat IL-1β, IL-6, and TNF-α -immunoassay kits (Abcam) were used to perform IL-1β, IL-6, and TNF-α quantifications. The 4.5 h solid-phase ELISA used in quantizing the rat IL-1β, IL-6, and TNF-α immunoassay was created to quantify the amounts of these three substances. An enzyme-linked immunosorbent test (ELISA) uses a solid-phase sandwich and a microtiter plate reader. From the standard curves, proinflammatory cytokine concentrations were determined.

The pre-coated plate was filled with the sample (100 μL). After that, the plate was sealed and left to sit at room temperature for two hours. With wash buffer, the plate was washed four times. Each well received 100 μL of diluted detection antibody solution, and the plate was then sealed and incubated at room temperature for 2 h. The plate was then cleaned with wash buffer four times. After adding 100 μL of diluted Streptavidin-HRP solution to each well, the plate was sealed and incubated at room temperature for 30 min. The plate was cleaned with a wash buffer four more times. Each well received 100 μL of diluted TMB substrate solution, and the plate underwent a 15 min incubation period in the dark. Positive wells then took on a bluish color. The reaction was stopped by adding 100 μL of stop solution to each well. Positive wells changed from blue to yellow. After halting the experiment, the absorbance at 450 nm was detected after 15 min.

### 4.7. Drug

The drug used in the study was melatonin, which was obtained from Sigma-Aldrich, USA. Melatonin was intraperitoneally administered (dose-dependently 30, 60, 120 mg/kg i.p.) [44].

### 4.8. Statistics

All data were analyzed using a Kruskal–Wallis test (a nonparametric analysis) followed by a Dunn post hoc test for multiple comparisons. * *p* < 0.05 was considered statistically significant.

## 5. Conclusions

Chronic melatonin administrations have been shown to alleviate pain behaviors in animal models of CPSP in various ways, such as by preserving mitochondrial homeostasis, reducing free radical generation, enhancing mitochondrial glutathione levels, and safeguarding the proton potential in the mitochondrial ETC by stimulating complex I and IV activities and reducing CPSP-induced neuroinflammation. Therefore, we suggested that administrations of exogenous melatonin could be promising clinical interventions in CPSP patients.

## Figures and Tables

**Figure 1 ijms-24-05413-f001:**
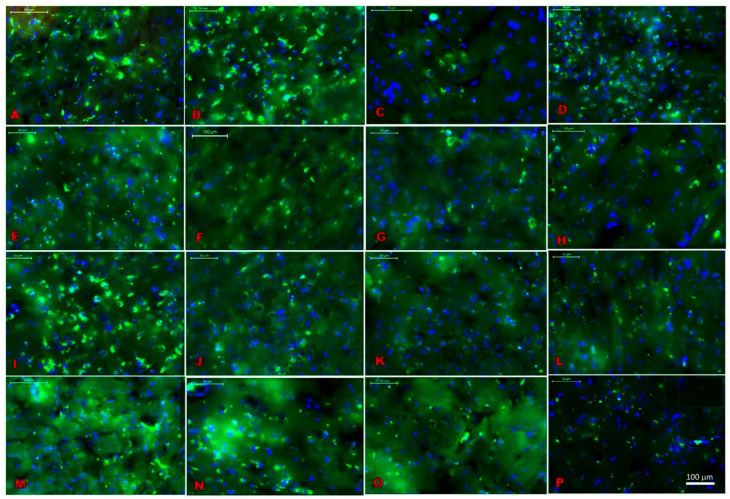
MT1 receptor distribution using FITC (florescent green) and DAPI (blue) staining in the specific brain regions. (**A**) Ventral posteromedial thalamic nucleus (VPM), (**B**) Somatosensory cortex (S1), (**C**) Dentate gyrus (DG), (**D**) Amygdalostriatal transition area (Astr), (**E**) Secondary auditory cortex (AUD), (**F**) Basolateral amygdala (Bla), (**G**) Lateral hypothalamus (LH), (**H**) Magnocellular region (Mc), (**I**) Magnocellular nucleus of lateral hypothalamus (Mclh), (**J**) Nigrostriatal fibers (Ns), (**K**) Posterior hypothalamic area (Ph), (**L**) Retrosplenial agranular cortex (Rsa), (**M**) Posterior thalamic nuclear group (Po), (**N**) Ventromedial thalamic nucleus (VM), (**O**) Ventromedial hypothalamic nucleus (VMH), and (**P**) Reticular thalamic nucleus (RT). Scale bar is 100 μm.

**Figure 2 ijms-24-05413-f002:**
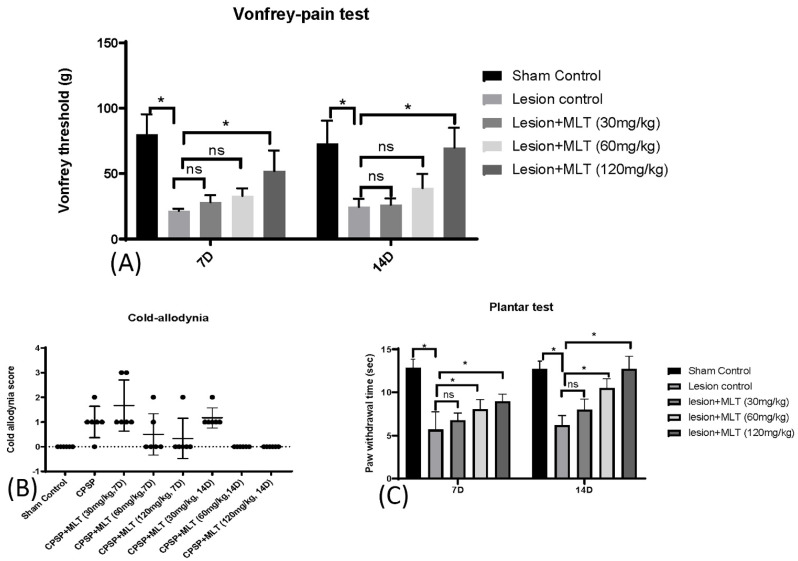
Effects of melatonin treatments on various pain tests, including Vonfrey pain, plantar, and cold allodynia tests. (**A**) Effect of melatonin administration on paw withdrawal threshold (g) in rats during a Von Frey pain test on days 7 and 14 after thalamic hemorrhage (n = 5, per group). Data expressed as mean ± SD. The figure shows pain hypersensitivity in central stroke rats 7 days and 14 days post-injury, * *p* < 0.05. (**B**) Effect of melatonin administration in cold allodynia in rats on days 7 and 14 after thalamic hemorrhage (n = 6, per group). The cold allodynia scores, as accounted by paw withdrawal, were demonstrated more in CPSP rats as compared with the sham control; MLT (30 mg/kg, i.p.) gave the same response as CPSP rats (not shown in figure); MLT (60 mg/kg i.p.) showed an effective response, but some rats showed both prolonged and quick paw withdrawal responses, whereas MLT (120 mg/kg i.p.) demonstrated a much better effect on CPSP rats. The cold allodynia score was almost zero on the 14th day of melatonin administration, which suggests a healing effect of melatonin. (**C**) Effect of melatonin administration on paw withdrawal latency in plantar test in rats on days 7 and 14 after thalamic hemorrhage (n = 5, per group). Data expressed as mean ± SD. The figure shows pain hyperalgesia in central stroke rats 7 days and 14 days post-injury, * *p* < 0.05.

**Figure 3 ijms-24-05413-f003:**
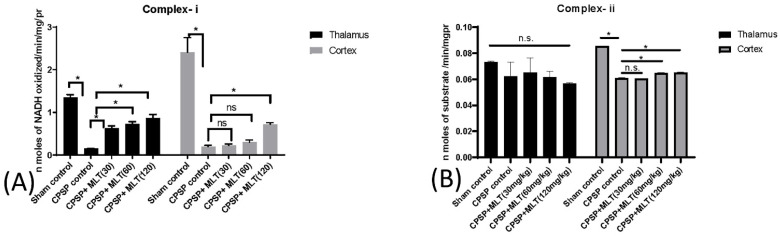
Effect of CPSP and melatonin treatment on mitochondria chain complexes, including (**A**) mitochondrial complex-i (NADH: ubiquinone oxidoreductase; n = 5, per group), (**B**) complex-ii (n = 4, per group), (**C**) complex-iii (n = 5, per group), and (**D**) complex-iv (n = 5, per group) levels in the thalamus and cortex after two weeks of melatonin administration. Data expressed as mean ± SD; * *p* < 0.005 as compared with CPSP.

**Figure 4 ijms-24-05413-f004:**
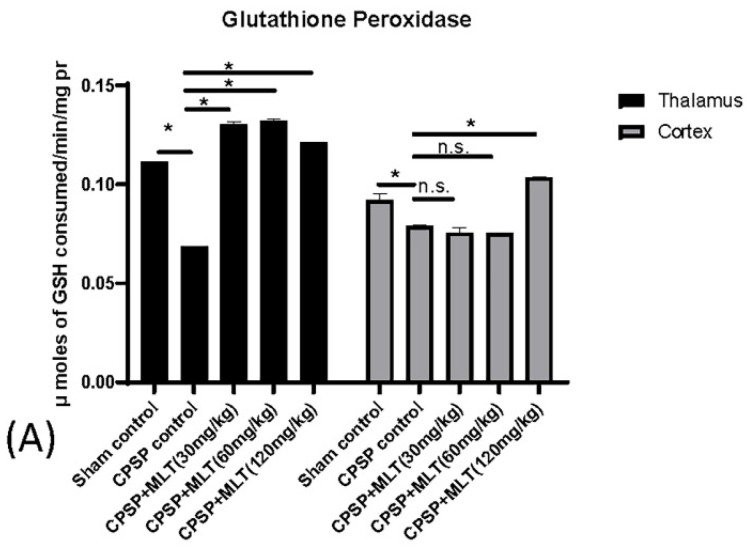
Effect of CPSP and melatonin on ETC chain enzymes A, including (**A**) gluthathione peroxidase (n = 4, per group), (**B**) catalase (n = 4, per group), and (**C**) superoxidase dismutase (n = 5, per group) levels in the thalamus and the cortex after two weeks of melatonin administration. Data expressed as mean ± SD; * *p* < 0.005 as compared with CPSP.

**Figure 5 ijms-24-05413-f005:**
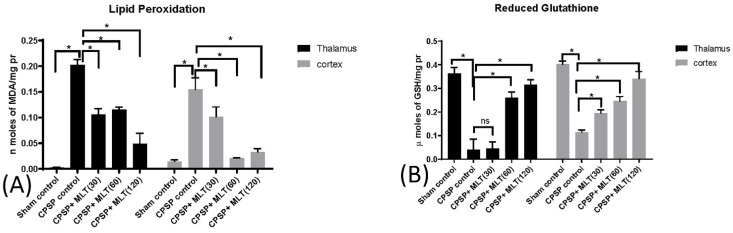
Effect of lesions and melatonin treatments on oxidative stress, including (**A**) lipid peroxidation (n = 5, per group) and (**B**) reduced glutathione (n = 5, per group) levels in the thalamus and the cortex after two weeks of melatonin administration. Data expressed as mean ± SD; * *p* < 0.005 as compared with CPSP.

**Figure 6 ijms-24-05413-f006:**
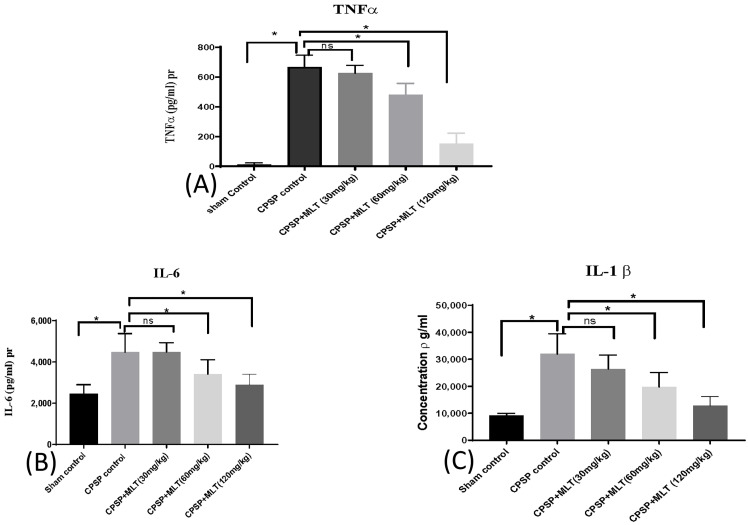
Effect of lesions and melatonin administration on neuroinflammation, including (**A**) TNF-a (n = 5, per group), (**B**) IL-6 (n = 6, per group), and (**C**) IL-1b (n = 6, per group) levels in the whole brain after two weeks of melatonin administration. Data expressed as mean ± SD; * *p* < 0.005 as compared with CPSP.

**Figure 7 ijms-24-05413-f007:**
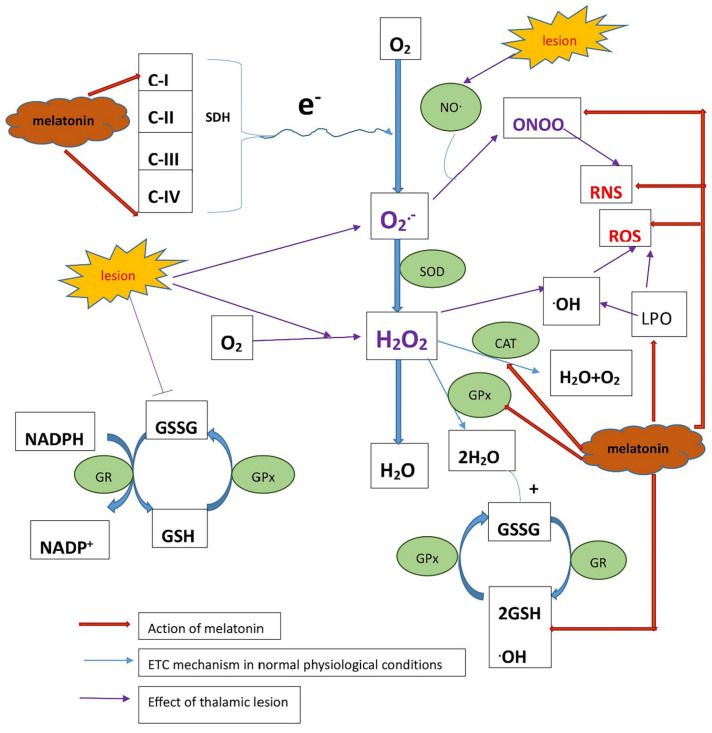
The schematic representation showing an ETC chain reaction in mitochondria in normal physiological conditions, effect of lesion, and effect of exogenous melatonin treatment on ETC. In normal conditions, the electron from the transport chain progresses to form water, but due to the lesion, the oxygen free radical produces nitric oxide radicals, giving rise to RNS. Mitochondrial superoxide dismutase (SOD) neutralizes the highly reactive superoxide radical O_2_–, known as reactive oxygen species (ROS). Under CPSP conditions due to the lesion, ROS and RNS levels increase and melatonin helps to reduce ROS and RNS.

**Figure 8 ijms-24-05413-f008:**
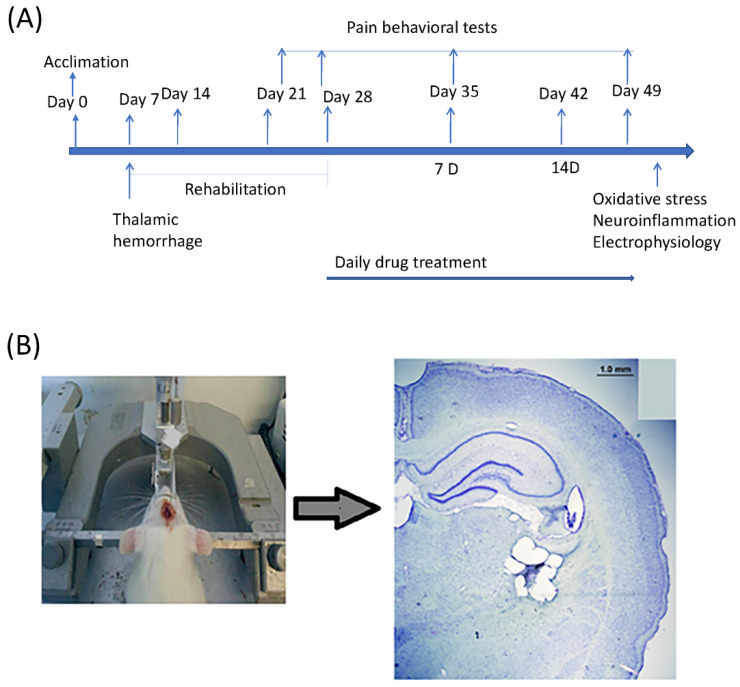
(**A**) Flowchart for experimental design. (**B**) Microinjection of collagenases to induce lesions of the VPL/VPM in the animal model of central post-stroke pain.

**Table 1 ijms-24-05413-t001:** Melatonin receptor distribution in regions of the brain.

Region of Interest	Annotation Area	Relative Mask Area	Total Count	Receptor Density
S1	7.67 mm^2^	0.12%	3170	413.1 mm^2^
VM	0.47 mm^2^	0.1%	211	449.45 mm^2^
RT	0.34 mm^2^	0.23%	265	768.52 mm^2^
Hippocampus	5.50 mm^2^	1.14%	4648	844.86 mm^2^
MEPD/MEPV	0.42 mm^2^	0.2%	415	996.89 mm^2^
VPM/VPL	2.26 mm^2^	0.88%	3274	1450.59 mm^2^
LH (lateral hypothalamus)	0.68 mm^2^	0.44%	592	867.58 mm^2^
Vmhyp nu (ventromedial hypothalamus)	2.12 mm^2^	0.09%	579	273.11 mm^2^
AUD	4.11 mm^2^	0.33%	3428	833.96 mm^2^
PH (posterior hypothalamus)	0.26 mm^2^	0.27%	163	635.45 mm^2^
BLA/BMP	0.73 mm^2^	0.08%	223	303.86 mm^2^
S2	1.01 mm^2^	0.23%	751	744.1 mm^2^

## Data Availability

The original contributions presented in the study are included in the article, further inquiries can be directed to the corresponding authors.

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
