# Peer review of "Modulation of Melatonin in Pain Behaviors Associated with Oxidative Stress and Neuroinflammation Responses in an Animal Model of Central Post-Stroke Pain"

_ijms, 2023, doi:10.3390/ijms24065413_

Round 1
Reviewer 1 Report
The manuscript by Kaur et al describes the modulatory effects of melatonin on pain behavior and underlying oxidative stress/neuroinflammatory response in the CPSP model.
Even though the study has some potential, there are substantial flaws in the experimental design.
1) The Authors did not state which brain structure was used for any of the analyses, leading this Reviewer to conclude that the whole brain was used. In the graphs, the Authors state that the cortex and thalamus were used. Which parts of the cortex and thalamus, how those structures were isolated? Which parts of the cortex? Why cortex was chosen? This is problematic because no clear conclusions can be drawn from the total fractions.
2) The Authors did not evaluate the purity of the mitochondrial fraction and many of the tested parameters may be found in other cell compartments.
3) The number of animals used for the behavior is too small to draw any conclusions.
4) The Authors did not test for Normality, however, with 5-6 samples per group Normality cannot be achieved which implies that parametric ANOVA was not used correctly.
5) The micrographs show highly non-specific background staining and no colocalization was made against markers of glial/neuronal cells but rather DAPI which stains microglia, neurons, astrocytes, pericytes, endothelial cells, oligodendrocytes, etc. Furthermore, it remains unclear why the Authors chose only MT1 receptors but not MT2.
6) The graphical representations are not unified, some graphs are shown as bars, and some are shown with a number of animals within.
7) It remains unclear how the authors showed the results of ELISA one graph for both structures? Did they mixed the samples?
8) English language needs extensive editing
Author Response
Reviewer 1:
The manuscript by Kaur et al describes the modulatory effects of melatonin on pain behavior and underlying oxidative stress/neuroinflammatory response in the CPSP model.
Even though the study has some potential, there are substantial flaws in the experimental design.
Point 1) The Authors did not state which brain structure was used for any of the analyses, leading this Reviewer to conclude that the whole brain was used. In the graphs, the Authors state that the cortex and thalamus were used. Which parts of the cortex and thalamus, how those structures were isolated? Which parts of the cortex? Why cortex was chosen? This is problematic because no clear conclusions can be drawn from the total fractions.
Response 1: Thank you for your comments. We isolated the whole thalamus and cortex, respectively, for the biochemical analysis to test the effect of cortex and thalamus separately in CPSP animals. After extraction of brain from the skull of rat, rinsed the brain with cold PBS to remove excess blood. Then isolated the tissues on dry ice with forceps and scissors. It was assessed by the whole cerebral cortex and whole thalamus. It is known that several cortical regions, such as the somatosensory cortex, prefrontal cortex, insular, and anterior cingulate cortex, are activated by acute pain signals, and neurons in these regions have been demonstrated to undergo changes in response to chronic pain, therefore we chosen cortex. Further, Role of thalamus in modulating pain is also well known. The relevant literatures were shown as follows. Also, we have added the description in the content of the manuscript. Please see Lines 432-442.
“…4.6. Experimental procedures for evaluating oxidative stress and mitochondrial function
4.6.1. Dissection and homogenization
After the behaviors, the animals were sacrificed. After brain extraction from the rat's skull, rinsed the brain with cold PBS to remove excess blood. Then isolate the tis-sues on dry ice with forceps and scissors. Because the previous studies demonstrated the thalamus and cerebral cortex contributed to chronic pain [31,32], the whole cerebral cortex and whole thalamus were determined for assessments. Later, brain samples were quickly taken out and put on dry ice to isolate the cortex and thalamus. 10% (w/v) tissue homogenates were then made in 0.1 M phosphate buffer (pH 7.4). The homogenates were centrifuged at 10,000 × g for 15 min. Aliquots of supernatants were separated and utilized for biochemical and molecular estimations…”
References :
Ab Aziz, Che Badariah, and Asma Hayati Ahmad. "The role of the thalamus in modulating pain." The Malaysian journal of medical sciences: MJMS 13.2 (2006): 11.
Xie, Yu-feng, Fu-quan Huo, and Jing-shi Tang. "Cerebral cortex modulation of pain." Acta Pharmacologica Sinica 30.1 (2009): 31-41.
Point 2) The Authors did not evaluate the purity of the mitochondrial fraction and many of the tested parameters may be found in other cell compartments.
Response 2: Thank you for your comments. We have assessed many mitochondria biomarkers including Complex-I, Complex-II, Complex-III, and Complex-IV. Particularly, the purity of the mitochondrial fraction that was assessed by the enriched activity of succinate dehydrogenase and it is also known as complex II. Please see the assessment of our complex II on Lines 505-512.
“…Complex - II: Also known as succinate dehydrogenase (SDH), was assayed with the modified method of King (1967) [39]. The various reagents used were: 100 ml of sodium phosphate buffer (0.2 M; pH 7.8), where NaH2PO4 = 3.12 gm in 100 ml and Na2HPO4 = 2.83 gm in 100 ml; BSA (1%) = 100 mg/100ml (unshaken); Succinic acid (freshly prepared) 0.6M-700 mg/10ml DDW; Potassium ferricyanide 0.03 M (freshly prepared), 9.8 mg/1 ml. The procedure was performed as follows: 200µl succinate was added to 1.5 ml of phosphate buffer, then added 300 µl BSA, 25 µl Pot. Ferricyanide, 1.75 ml DDW and 25 µl sample and Check change in OD at 420 for 180 min…”
Point 3) The number of animals used for the behavior is too small to draw any conclusions.
Response 3: Thank you for your comments. For every group, we used n=6 number of animals. In the animal behavioral model, n = 6 number of animals are fine if the significant differences were occurred. Thus, we found that our study just used n = 6 is ok.
Point 4) The Authors did not test for Normality, however, with 5-6 samples per group Normality cannot be achieved which implies that parametric ANOVA was not used correctly.
Response 4: Thank you for your comments. Based on the rule of the statistical analysis, the normality is one of the assumptions of the ANOVA. Before analyzing the data with ANOVA, we will assume the distribution of data is normal distribution. Thus, it will not be asked to test the normality.
Point 5) The micrographs show highly non-specific background staining and no colocalization was made against markers of glial/neuronal cells but rather DAPI which stains microglia, neurons, astrocytes, pericytes, endothelial cells, oligodendrocytes, etc. Furthermore, it remains unclear why the Authors chose only MT1 receptors but not MT2.
Response 5: Thank you for your comments. Recent research indicates that MT1 receptor is involved in neural pathways modulating depression and diurnal rhythms. Therefore, MT1 may play an important role in the signaling pathway transduction of the nervous system. Several studies indicate that the MT1, rather than MT2, receptor is implicated in circadian rhythm regulation (see Comai et al., 2015). And our previous study indicated that there is a bidirectional link between Pain and circadian rhythms (Kaur et al., 2022). Therefore, we studied MT1 in pain model of CPSP. Please see the following papers.
References:
Kaur, Tavleen, et al. "Modulation of melatonin to the thalamic lesion-induced pain and comorbid sleep disturbance in the animal model of the central post-stroke hemorrhage." Molecular Pain 18 (2022): 17448069221127180.
Comai, Stefano, et al. "Melancholic-like behaviors and circadian neurobiological abnormalities in melatonin MT1 receptor knockout mice." International Journal of Neuropsychopharmacology 18.3 (2015).
Point 6) The graphical representations are not unified, some graphs are shown as bars, and some are shown with a number of animals within.
Response 6: Thank you for your comments. All graphs are bar graphs, except cold allodynia, which represents the scores of paw withdrawal responses. That is because the data of cold allodynia showed a big variance and the data were discrete variable but not the continuous variable. However, the behavioral data of the other figures were continuous variable. Thus, the cold allodynia data were not suitable for bars.
Point 7) It remains unclear how the authors showed the results of ELISA one graph for both structures? Did they mixed the samples?
Response 7: Thank you for your comments. We did not mix any samples. We performed biochemical estimations separately, and then plotted graphs for both cortex and thalamus samples, respectively.
Point 8) English language needs extensive editing
Response 8: Thank you for your comments. The English quality has been improved by a native English editor for the whole article.

Reviewer 2 Report
1. Figure 1 is not fully visible, but I suppose that is a matter of editing.
2. Data in Table 1 should be presented as mean ± SD from at least three samples.
3. ROS level in the thalamus and cortex should be evaluated.
4. Why levels of inflammatory cytokines (Figure 7) were measured in whole brain, not separated as in other figures?
5. Effects of melatonin on signaling pathways related to oxidative stress and inflammation (for example, MAPK, NF-kappaB) should be evaluated.
Author Response
Reviewer 2: Comments and Suggestions for Authors
Point 1: Figure 1 is not fully visible, but I suppose that is a matter of editing.
Response 1: Thank you for your comments. We have followed your comments to modify Figure 1. The Figure1 has been visible as possible as we can. Please see Figure 1.
Point 2: Data in Table 1 should be presented as mean ± SD from at least three samples.
Response 2: Thank you for your comment. Because we use the statistical analysis method to perform a random sampling way for one immunofluorescence in each specific brain area, we cannot show the average and standard deviation for the determined brain areas. Thus, Table 1 did not show a mean + SD for each brain area. However, we choose to use the quality analysis, not quantitative, in Figure 1 and Table 1. Thank you for your suggestion.
Point 3: ROS level in the thalamus and cortex should be evaluated.
Response 3: Thank you for your comments. ROS are highly reactive molecules and are extremely unstable, so detection of ROS relies on measuring the end products that are formed when they react with particular substances. The end products can be measured by fluorescence, color, or luminescence changes. For instance, the main effect of ROS is lipid peroxidation (Figure 5), which occurs when membrane phospholipids are brought into contact with a ROS oxidizing agent, and Lipid peroxidation, which forms a pink color adduct. Thus, because of our study's high variance for ROS levels, we did not evaluate ROS levels. Instead, we measured lipid peroxidation and reduced glutathione levels in the thalamus and cortex to replace the evaluation of ROS levels.
Point 4: Why levels of inflammatory cytokines (Figure 7) were measured in whole brain, not separated as in other figures?
Response 4: Thank you for your comments. There were two reasons to explain why we used the whole brain for assessing neuroinflammation in the present study. First of all, for neuroinflammation responses, we used the whole brain as neuroinflammation, even when local, with a high chance that it might progress into disseminated inflammation through infiltration of inflammatory mediators. Second, the purpose of the study focuses on the whole brain’s neuroinflammation responses after destroying the VPL and VPM complex induced central post-stroke pain. The specific brain area is not our key point in the study. Thank you for your understanding.
Point 5: Effects of melatonin on signaling pathways related to oxidative stress and inflammation (for example, MAPK, NF-kappaB) should be evaluated.
Response 5: Thank you for your comments. Although the current study did not examine how melatonin inhibits the NF-κB and MAPK pathways resulting in changes in central post-stroke pain, we have added this issue in the Discussion section. Please see Lines 349-352.
“…In summary, how melatonin, through the signaling pathways of MAPK or NFκB mod-ulates oxidative stress and inflammation due to changes in central post-stroke pain should be evaluated in further studies…”

Round 2
Reviewer 1 Report
The Authors did address some of the concerns, however, the severe problems regarding the design and statistics still stand.
You cannot assume Normality and use parametric ANOVA, without checking the normality (there are tests in virtually every program you can use). However, with 5-6 samples per group, you cannot assume normality.
Furthermore, six animals per group in the experimental design with 4+ groups is not enough to demonstrate the phenomenon with certainty.
The Authors did not address the problems with high background staining and why the Authors did not use any specific markers.
Author Response
Reviewer 1: Comments and Suggestions for Authors
The Authors did address some of the concerns, however, the severe problems regarding the design and statistics still stand.
Point 1: You cannot assume Normality and use parametric ANOVA, without checking the normality (there are tests in virtually every program you can use). However, with 5-6 samples per group, you cannot assume normality.
Response 1: Thank you for your comment. Because our samples are fewer about n = 5-6 per group. Accordingly, the normality test was significant differences for all ANOVA analysis. Therefore, we used Kruskal-Wallis test (Nonparametric analysis) and Post hoc with Dunn test to replace the ANOVA analysis and Post hoc Tukey tests. Please see the Result section.
Point 2: Furthermore, six animals per group in the experimental design with 4+ groups is not enough to demonstrate the phenomenon with certainty.
Response 2: Thank you for your comment. Because the samples were n = 5-6 per group, we used Kruskal-Wallis test (Nonparametric analysis) and Post hoc with Dunn test to replace the ANOVA analysis and Post hoc Tukey tests to demonstrate the phenomenon with certainty. Please see the Result section.
Point 3: The Authors did not address the problems with high background staining and why the Authors did not use any specific markers.
Response 3: Thank you for your comment. We used MT1 specific antibody for the staining. Due to the longer exposure time for taking the image, caused high background staining, and this was the best quality image we could achieved. However, the binding signal is positive. Now we have readjusted the parameter of signal/noise ratio and reduced the background brightness of the image. We did not use any other specific markers as we only aimed to quantify the expression levels of melatonin receptors (MT1) in the various regions of brain including thalamus and cortex. We did not focus on discriminating expression levels in particular cell types, so we didn’t target specific cell types with other specific markers.

Reviewer 2 Report
The revised version of the manuscript is acceptable for publication.
Author Response
Dear Reviewer 2,
Thank you.
Andrew

Round 3
Reviewer 1 Report
The Authors addressed all the main concerns of this Reviewer.